# POSTHOC PRIVACY GUARANTEES FOR NEURAL NETWORK QUERIES

## ABSTRACT

Cloud based machine learning inference is an emerging paradigm where users share their data with a service provider. Due to increased concerns over data privacy, recent works have proposed using Adversarial Representation Learning (ARL) to learn a privacy-preserving encoding of sensitive user data before it is shared with an untrusted service provider. Traditionally, the privacy of these encodings is evaluated empirically as they lack formal guarantees. In this work, we develop a new framework that provides formal privacy guarantees for an arbitrarily trained neural network by linking its local Lipschitz constant with its local sensitivity. To utilize local sensitivity for guaranteeing privacy, we extend the Propose-Test-Release (PTR) framework to make it tractable for neural network based queries. We verify the efficacy of our framework experimentally on real-world datasets and elucidate the role of ARL in improving the privacy-utility tradeoff.

## 1 INTRODUCTION

The ethical and regulatory concerns around data privacy have become increasingly important with the adoption of machine learning (ML) across various sectors such as health, finance, and mobility. Although training ML models privately has seen tremendous progress(Abadi et al. (2016); Papernot et al. (2016); Du et al. (2021); Jordon et al. (2018)) in the last few years, protecting privacy during the inference phase remains a challenge as these models get deployed by cloud based service providers. Cryptographic techniques(Ohrimenko et al. (2016); Knott et al. (2021); Mishra et al. (2020); Juvekar et al. (2018)) address this challenge by performing computation over encrypted data. However, to combat the high computational cost of encryption techniques, alternative works have used ARL to suppress task irrelevant information from data. While ARL based techniques have shown promising empirical results, they lack formal privacy guarantees over obfuscated representations due to their use of Deep Neural Networks (DNNs) for achieving privacy. For the first time, we show how to give formal privacy guarantees for inference queries over arbitrarily trained (including ARL) DNNs.

The key aspect of any ARL algorithm is an *obfuscator* which is trained to encode a user's private data such that an attacker can not recover the original data from its encoding. Achieving formal privacy guarantees for an *obfuscator* has remained elusive due to the non-convexity of the training objective of DNNs. In this work, we take a *posthoc* approach to guaranteeing privacy, where the privacy of data is evaluated after the *obfuscator* is learned. Because the *obfuscator* is trained for non-invertibility, we hypothesize that the *obfuscator* network should act as a contractive mapping, and hence, increase the stability of the function in its local neighborhood, i.e., reduce sensitivity. Therefore, we measure the stability of an adversarially learned *obfuscator* neural network, using Lipschitz constants, and link it with privacy properties. To exactly compute the local Lipschitz constant of a non-linear (ReLU) DNNs, we use LipMip(Jordan & Dimakis (2020)), a mixed-integer programming based technique, and re-formulate the ARL pipeline to ensure the computational feasibility of calculating the Lipschitz constant. To draw a connection between the local Lipschitz constant and reconstruction privacy, we introduce a privacy definition that is a specific instantiation of a general $d_\chi$-privacy framework by Chatzikokolakis et al. (2013). Instead of evaluating the global Lipschitz constant of DNNs, we evaluate the Lipschitz constant only in the local neighborhood of the user's sensitive data. We extend the Propose-Test-Release (PTR)(Dwork & Lei (2009)) framework to formalize our local neighborhood based measurement of the Lipschitz constant.

The scope of our paper is to provide privacy guarantees against reconstruction attacks for existing ARL techniques, i.e., our goal is not to develop a new ARL technique but rather to develop a formal privacy framework compatible with existing ARL techniques. A Majority of the ARL techniques protect either a sensitive attribute or reconstruction of the input. We only consider sensitive input in this work. We adopt a different threat model from that of traditional differential privacy (DP)(Dwork et al. (2014)) because, as we explain later, protecting membership inference is at odds with private inference. Our threat model for the reconstruction attack is motivated by the use cases where a user may be willing to disclose coarse-grained information about their data but wants to prevent leakage of fine-grained information. Alternate threat models have been widely used in the privacy literature(Chatzikokolakis et al. (2013); Kifer & Machanavajjhala (2014); Andrés et al. (2013); Hannun et al. (2021)). Furthermore, we only focus on protecting the privacy of data during the inference stage, and assume that ML models can be trained privately.

Typically ARL techniques evaluate the privacy of their representations by empirically measuring the information leakage using a proxy adversary. Existing works(Srivastava et al. (2019); Guo et al. (2021); Singh et al. (2021)) show that a proxy adversary's performance as a measure of protection could be unreliable. Some of the existing ARL techniques have used theoretical tools(Hamm (2017); Zhao et al. (2020b); Basciftci et al. (2016); Zhao et al. (2020a); Wang et al. (2017); Bertran et al. (2019); Mireshghallah et al. (2021)) for measuring information leakage empirically. However, most of these works analyze specific obfuscation techniques and lack formal privacy definitions. In contrast, our work is agnostic to the design of the *obfuscator* as long as it is differentiable, and our definition is built upon a variant of DP – a widely used formal privacy framework. Our privacy definition and mechanism is built upon $d_\chi$-privacy(Chatzikokolakis et al. (2013)) and PTR(Dwork & Lei (2009)). Existing instantiations of $d_\chi$-privacy include geo-indistinguishability(Andrés et al. (2013)) and location-dependent privacy(Koufogiannis & Pappas (2016)) that share a similar goal as ours of sharing coarse grained information. Our work differs in its usage of neural network queries and high dimensional data modality. We refer the reader to Appendix Sec A for a detailed literature review.

In Sec 2 we begin with the preliminaries of DP and its variant for metric spaces. Then, we motivate our ML inference setup and introduce our privacy definition in Sec 3. Next, we construct our posthoc framework by extending PTR and proving its privacy guarantees in Sec 4. In Sec 5 we experimentally demonstrate the feasibility of our framework and understand the dynamics of ARL algorithms. Our contributions can be summarized as follows:

- We introduce $(\epsilon, \delta, R)$-neighborhood privacy definition to formalize reconstruction privacy for ARL based inference.

- We extend the PTR framework to make it tractable for neural network based queries. Our extension bridges the gap between formal privacy frameworks and empirical techniques in private ML inference.

- We perform extensive experimental analysis on ARL techniques and provide insight into how ARL improves the privacy-utility tradeoff by reducing the local sensitivity of DNNs.

## 2    PRELIMINARIES

Differential privacy (DP)(Dwork et al. (2014)) is a widely used framework for answering a query, $f$, on a dataset $\mathbf{x} \in \chi$ by applying a mechanism $\mathcal{M}(\cdot)$ such that the probability distribution of the output of the mechanism $\mathcal{M}(\mathbf{x})$ is *similar* regardless the presence or absence of any individual in the dataset $\mathbf{x}$. More formally, $\mathcal{M}$ satisfies $(\epsilon, \delta)$-DP if $\forall \mathbf{x}, \mathbf{x}' \in \chi$ such that $d_H(\mathbf{x}, \mathbf{x}') \leq 1$, and for all (measurable) output $S$ over the range of $\mathcal{M}$

$$\mathbb{P}(\mathcal{M}(\mathbf{x}) \in S) \leq e^\epsilon \mathbb{P}(\mathcal{M}(\mathbf{x}') \in S) + \delta,$$

where $d_H$ is the hamming distance. This definition is based on a trusted central server model, where a trusted third party collects sensitive data and computes $\mathcal{M}(\mathbf{x})$ to share with untrusted parties. In *local*-DP(Kasiviswanathan et al. (2011)), this model has been extended such that each user shares $\mathcal{M}(\mathbf{x})$, and the service provider is untrusted. Our threat model is a special case of local DP which we refer to as **single-instance sharing**. In this setup, the client queries every data instance independently with the service provider and there is no aggregation or summary statistic involved. For ex.– a user shares a face image to receive an age prediction from the service provider. While our setup

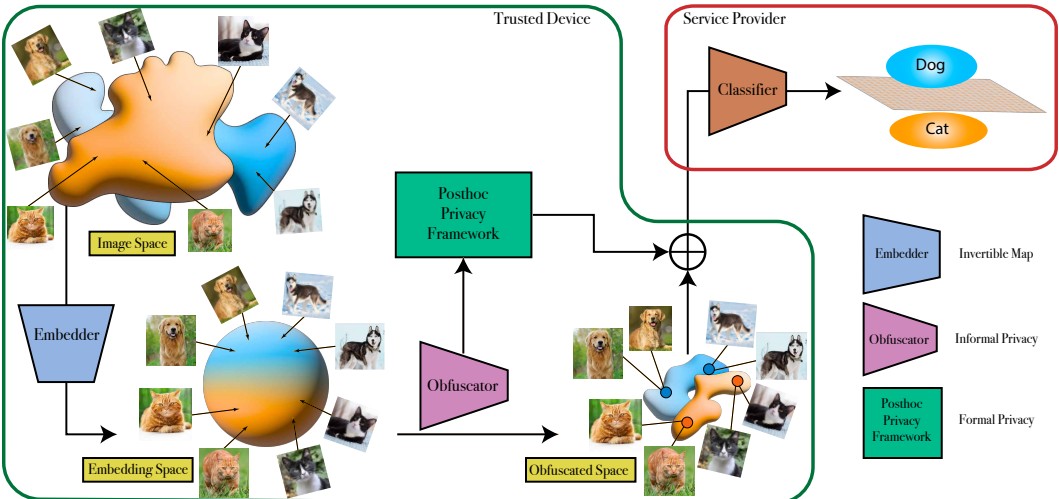

Figure 1: **Posthoc Privacy framework**: We project a high dimensional data instance to a lower dimensional embedding. The goal of the *embedder* is to measure a semantically relevant distance between different instances. The embedding is fed to the *Obfuscator* that compresses similar inputs in a small volume. In traditional ARL, the obfuscated instance is shared with the untrusted service provider without any formal privacy guarantee. In this work, by analyzing the stability of the obfuscator network, we perturb the obfuscated instance to provide a formal privacy guarantee.

is similar to item-level local DP, the answer to the query depends exactly on a single input. We note that $d_H(\mathbf{x}, \mathbf{x}') \leq 1$, $\forall \mathbf{x}, \mathbf{x}' \in \mathcal{X}$, whenever single-instance sharing is involved. Informally, this notion of neighboring databases under the DP definition would suggest that the outcome of two individuals should be *similar* no matter how different their datum is. This privacy definition could be too restrictive for our ML inference application where the data instance necessarily needs a certain degree of distinguishability to obtain utility from the service provider. This observation is formalized in the impossibility result of instance encoding(Carlini et al. (2020)) for private learning. To subside this fundamental conflict between the privacy definition and our application, we look at the definition of $d_\chi$-privacy(Chatzikokolakis et al. (2013)) that generalizes the DP definition to a general distance metric beyond hamming distance as follows:

$$\mathbb{P}(\mathcal{M}(\mathbf{x}) \in S) \leq e^{d_\chi(\mathbf{x}, \mathbf{x}')}\mathbb{P}(\mathcal{M}(\mathbf{x}') \in S), \tag{1}$$

here $d_\chi(\mathbf{x}, \mathbf{x}')$ is a function that gives a level of indistinguishability between two datasets $\mathbf{x}$ and $\mathbf{x}'$. DP can be viewed as a special case of $d_\chi$-privacy by keeping $d_\chi(\mathbf{x}, \mathbf{x}') = \epsilon d_H(\mathbf{x}, \mathbf{x}')$. Choosing a different distance metric yields stronger or weaker privacy guarantee.

## 3 PRIVACY DEFINITION

In order to formalize reconstruction privacy, we hypothesize that semantically similar points are close to each other on a data manifold, i.e. semantically similar samples are closer in a space where distances are defined as geodesic on the manifold. Therefore, one way to bound the reconstruction of $\mathbf{x}$ is by making it indistinguishable among semantically similar points. The extent of reconstruction privacy would therefore depend upon the radius of the neighborhood. We formalize it by introducing a privacy parameter $R$ that allows a user to control how big this indistinguishable neighborhood should be. This formulation leads to two additional constraints - i) a distance metric that models low dimensional manifold space of data; ii) a privacy definition that incorporates the privacy parameter $R$ as well as the distance metric. We propose to use embedding based manifold learning techniques(Brehmer & Cranmer (2020); Horvat & Pfister (2021)) for the first constraint because we do not have a closed form expression for the *manifold chart* for real world data. We refer to the distance metric as $d_\theta^\beta(\mathbf{x}, \mathbf{x}')$, where the parameter $\theta$ is learned to model the data manifold and $\beta$ is a standard norm such as $\ell_1, \ell_2$. Intuitively, we want to compute distances in a space where semantically similar data points are closer and semantically different data points are farther apart. For high dimensional datasets that lie

over a low dimensional manifold (such as images), traditional distance metrics like $\ell_1, \ell_2$ norms do not capture the semantic similarity. This idea has been used in perceptual similarity for computer vision(Zhang et al. (2018)) as well as manifold and metric learning techniques(Brehmer & Cranmer (2020); Horvat & Pfister (2021); Kha Vu (2021)). Hence, we instantiate $d_\chi$-privacy by keeping $d_\chi(\mathbf{x}, \mathbf{x}') = \epsilon d_\theta^\beta(\mathbf{x}, \mathbf{x}')$ such that,

$$\mathbb{P}(\mathcal{M}(\mathbf{x}) \in S) \leq e^{\epsilon d_\theta^\beta(\mathbf{x}, \mathbf{x}')} \mathbb{P}(\mathcal{M}(\mathbf{x}') \in S) + \delta. \tag{2}$$

The privacy parameter $\epsilon$ describes the extent of indistinguishability and the parameter $R$ describes the neighborhood in which we obtain this indistinguishability. We note that $d_\chi$-privacy, unlike DP, does not use the notion of a neighborhood ($d_\theta^\beta(\mathbf{x}, \mathbf{x}') \leq R$) because the guarantee holds for any possible pair of $\mathbf{x}, \mathbf{x}' \in \chi$ and smoothly decays with distance. Finally, we slightly weaken the $d_\chi$-privacy instantiation by defining neighborhood as $d_\theta^\beta(\mathbf{x}, \mathbf{x}') \leq R$ and keeping fixed levels of indistinguishability $d_\theta^\beta(\mathbf{x}, \mathbf{x}') \leq R$.

**Definition 1.** *A mechanism $\mathcal{M}$ satisfies $(\epsilon, \delta, R)$-semantic neighborhood privacy if $\forall \mathbf{x}, \mathbf{x}' \in \chi$ s.t. $d_\theta^\beta(\mathbf{x}, \mathbf{x}') \leq R$ and $S \subseteq Range(\mathcal{M})$*

$$\mathbb{P}(\mathcal{M}(\mathbf{x}) \in S) \leq e^\epsilon \mathbb{P}(\mathcal{M}(\mathbf{x}') \in S) + \delta. \tag{3}$$

Note that the above equation is exactly the same as $(\epsilon, \delta)$-DP except for the definition of neighboring databases. In this way, our privacy definition can be seen as a mix of standard DP and $d_\chi$-prviacy. A key characteristic of Eq 2 is that points closer to a given $\mathbf{x}$ than $R$ enjoy higher indistinguishability while in Eq 3 all points in the neighborhood of $\mathbf{x}$, similar to DP, get the same level of indistinguishability.

### 3.1 COMPARISON WITH DIFFERENTIAL PRIVACY

Conceptually, usage of hamming distance in DP for neighboring databases provides a level of protection such that the output does not change *significantly* regardless of the chosen sample. Such a privacy requirement can be at odds with the goal of prediction that necessarily requires discrimination between samples belonging to different concept classes. Our privacy definition relaxes this dichotomy by using a distance metric in the embedding space and guarantees privacy only within a neighborhood. The size of the neighborhood is a privacy parameter $R$ such that higher value of $R$ provides higher privacy. This privacy parameter is equivalent to the *group size*(Dwork et al. (2014)) used sometimes in the DP literature. By default, this value is kept 1 in DP but can be kept higher if a group of individuals (family, community) have to be privatized instead of a single individual. There is an equivalence between the group privacy definition and standard DP definition which we state informally -
**Lemma 2.2** in Vadhan (2017): Any $(\epsilon, \delta)$-differentially private mechanism is $(R\epsilon, Re^{(R-1)\epsilon}\delta)$-differentially private for groups of size $R$.
This lemma also applies to our proposed definition. However, we emphasize that privacy parameters of $(\epsilon, \delta)$-DP mechanism can not be compared trivially with a $(\epsilon, \delta, R)$-semantic neighborhood privacy mechanism because same value of $\epsilon$ and $\delta$ provide different levels of protection due to different definitions of neighboring databases. We experimentally demonstrate this claim in Sec 5.

## 4 PRIVACY MECHANISM

Our goal is to design a framework that can provide a formal privacy guarantee for single data instance sharing that is informally privatized using ARL. However, ARL algorithms use non-linear neural networks trained on non-convex objectives making it difficult to perform any worst-case analysis. Therefore, we take a posthoc approach where the formal privatization is performed after the model is trained. Specifically, we apply propose-test-release (PTR) mechanism by Dwork & Lei (2009). Applying PTR directly to our query (ARL) is not computationally feasible because PTR requires estimating local sensitivity at multiple points whereas evaluating local sensitivity of a neural network query is not even feasible at a single point. Therefore, we design a tractable variant of PTR that utilizes local lipschitz constant estimator to compute privacy related parameters. We refer the reader to Appendix for a detailed discussion on the lipschitz constant estimation and PTR.

Conventionally, ARL algorithms have three computational blocks during the training stage: 1) *obfuscator* ($f(\cdot)$) that generates a (informally private) representation ($\bar{\mathbf{z}}$) of data, 2) *proxy adversary*

that reconstructs the data from the representation produced by the *obfuscator*, and 3) *classifier* that performs the given task using the obfuscated representation. The *classifier* and *proxy adversary* are trained to minimize the task loss and reconstruction loss, respectively. The *obfuscator* is trained to minimize the task loss but maximize the reconstruction loss. This setup results in a min-max optimization where the trade-off between task performance and reconstruction is controlled by a hyper-parameter $\alpha$. Note that some techniques(Oh et al. (2016); Osia et al. (2020); Vepakomma et al. (2021)) do not require a proxy adversary but still learn an obfuscator model using other regularizers. We propose to use an embedder $(g(\theta, \cdot))$ to *learn* semantic similarity using VAE(Kingma & Welling (2013)). The key idea of using the embedder is to embed the original sample $(\mathbf{x})$ to a lower dimensional space $(\mathbf{z} = g(\mathbf{x}))$ such that the distance metric in $\mathbf{z}$ space captures semantic similarity as shown in Fig 1 and motivated in Sec 3. Since $\mathbf{z}$ can be (almost) loss-lessly inverted to $\mathbf{x}$, it is fed to the *obfuscator* to get $\tilde{\mathbf{z}} = f(\mathbf{z})$.

Our framework applies the mechanism $\mathcal{M}$ such that the final released data $\hat{\mathbf{z}} = \mathcal{M}(\mathbf{x})$ has a privacy guarantee discussed in Eq 3. Like PTR, we start with a proposal $(\Delta_{LS}^{p})$ on the upper bound of the local sensitivity of $\mathbf{x}$. To test the validity of $\Delta_{LS}^{p}$, we compute the size of the biggest possible neighborhood such that the local Lipschitz constant of the *obfuscator* network in the neighborhood is lesser than the proposed bound. Next, we privately verify the correctness of the proposed bound for the given data instance. We do not release the data (denoted by $\perp$) if the proposed bound is invalid. Otherwise, we perturb the data using Laplace distribution calibrated by the proposed bound. Next, we discuss the framework and privacy guarantees in more detail.

**Global Sensitivity and Lipschitz constant** of a query $f : \mathcal{X} \rightarrow \mathcal{Y}$ are essentially same in the $d_{\mathcal{X}}$-privacy framework. Global sensitivity of a query $f(\cdot)$ is the smallest value of $\Delta$ (if it exists) such that $\forall \mathbf{x}, \mathbf{x}' \in \mathcal{X}, d_{\mathcal{Y}}(f(\mathbf{x}), f(\mathbf{x}')) \leq \Delta d_{\mathcal{X}}(\mathbf{x}, \mathbf{x}')$. While global sensitivity is a measure over all possible pairs of data in the data domain $\mathcal{X}$, local sensitivity $(\Delta_{LS})$ is defined with respect to a given dataset $\mathbf{x}$ such that $\forall \mathbf{x}' \in \mathcal{X}, d_{\mathcal{Y}}(f(\mathbf{x}), f(\mathbf{x}')) \leq \Delta_{LS}(\mathbf{x}) d_{\mathcal{X}}(\mathbf{x}, \mathbf{x}')$. We integrate the notion of semantic similarity in a neighborhood (described in Sec 2) by defining the local sensitivity of a neighborhood $\mathcal{N}(\mathbf{x}, R)$ around $\mathbf{x}$ of radius $R$ such that $\mathcal{N}(\mathbf{x}, R) = \{\mathbf{x}' | d_{\mathcal{X}}(\mathbf{x}, \mathbf{x}') \leq R, \forall \mathbf{x}' \in \mathcal{X}\}$. Therefore, the local sensitivity of query $f$ on $\mathbf{x}$ in the $R$-neighborhood is defined $\forall \mathbf{x}' \in \mathcal{N}(\mathbf{x}, R)$ such that

$$d_{\mathcal{Y}}(f(\mathbf{x}), f(\mathbf{x}')) \leq \Delta_{LS}(\mathbf{x}, R) d_{\mathcal{X}}(\mathbf{x}, \mathbf{x}'). \tag{4}$$

We note that if $d_{\mathcal{X}}$ is hamming distance and $R$ is 1 then this formulation is exactly same as local sensitivity in $\epsilon$-DP(Dwork et al. (2014)). The equation above can be re-written as:

$$\Delta_{LS}(\mathbf{x}, R) = \sup_{\mathbf{x}' \in \mathcal{N}(\mathbf{x}, R)} \frac{d_{\mathcal{Y}}(f(\mathbf{x}), f(\mathbf{x}'))}{d_{\mathcal{X}}(\mathbf{x}, \mathbf{x}')}. \tag{5}$$

This formulation of local sensitivity is similar to the definition of the local Lipschitz constant. The local Lipschitz constant $\mathcal{L}$ of $f$ for a given open neighborhood $\mathcal{N} \subseteq \mathcal{X}$ is defined as follows:

$$\mathcal{L}^{\alpha,\beta}(f, \mathcal{N}) = \sup_{\mathbf{x}', \mathbf{x}'' \in \mathcal{N}} \frac{||f(\mathbf{x}') - f(\mathbf{x}'')||_{\alpha}}{||\mathbf{x}' - \mathbf{x}''||_{\beta}} \quad (\mathbf{x}' \neq \mathbf{x}'') \tag{6}$$

We note that while the local sensitivity of $\mathbf{x}$ is described around the neighborhood of $\mathbf{x}$, the Lipschitz constant is defined for every possible pair of points in a given neighborhood. Therefore, in Lemma 4.1 we show that the local Lipschitz in the neighborhood of $\mathbf{x}$ is an upper bound on the local sensitivity.

**Lemma 4.1.** *For a given $f$ and for $d_{\mathcal{Y}} \leftarrow \ell_{\alpha}$ and $d_{\mathcal{X}} \leftarrow \ell_{\beta}$, $\Delta_{LS}(\mathbf{x}) \leq \mathcal{L}(f, \mathcal{N}(\mathbf{x}, R))$. Proof in Appendix B.1.*

Since local sensitivity is upper bounded by the Lipschitz constant, evaluating the Lipschitz constant suffices as an alternative to evaluating local sensitivity.

**Lower bound on testing the validity of $\Delta_{LS}^{p}$:** The PTR algorithm(Dwork & Lei (2009)) suggests a proposal on the upper bound $(\Delta_{LS}^{p})$ of local sensitivity and then finds the distance between the given dataset $(\mathbf{x})$ and the closest dataset for which the proposed upper bound is not valid. Let $\gamma(\cdot)$ be a distance query and $\Delta_{LS}(\mathbf{x})$ be the local sensitivity defined as per the DP framework with respect to $\mathbf{x}$ such that

$$\gamma(\mathbf{x}) = \min_{\mathbf{x}' \in \mathcal{X}} \{d_{H}(\mathbf{x}, \mathbf{x}') \ s.t. \ \Delta_{LS}(\mathbf{x}') > \Delta_{LS}^{p}\}. \tag{7}$$

In our framework, the query $\gamma(\mathbf{x}, R)$ can be formulated in the semantic neighborhood as follows:

$$\gamma(\mathbf{x}, R) = \min_{\mathbf{x}' \in \mathcal{X}} \{d_{\mathcal{X}}(\mathbf{x}, \mathbf{x}') \ s.t. \ \Delta_{LS}(\mathbf{x}', R) > \Delta_{LS}^{p}\}. \tag{8}$$

We note that keeping $d_\chi = d_H$ and $R = 1$, makes the $\gamma$ query exactly same as defined in the eq 7. In our setup, computing $\gamma(\cdot)$ is intractable due to local sensitivity estimation required for every $\mathbf{x}' \in \chi$ (which depends upon a non-linear neural network). We emphasize that this step is intractable at two levels, first we require estimating local sensitivity of a neural network query. Second, we require this local sensitivity over all samples in the data domain. Therefore, we make it tractable by computing a lower bound over $\gamma(\mathbf{x}, R)$ by designing a function $\phi(\cdot)$ s.t. $\phi(\mathbf{x}, R) \leq \gamma(\mathbf{x}, R)$. Intuitively, $\phi(\cdot)$ finds the largest possible neighborhood around $\mathbf{x}$ such that the local Lipschitz constant of the neighborhood is smaller than the proposed local sensitivity. Because the subset of points around $\mathbf{x}$ whose neighborhood does not violate $\Delta_{LS}^p$ is half of the size of the original neighborhood in the worst case, we return half of the size of neighborhood as the output. We describe its computation in Algorithm 1. More formally,

$$\phi(\mathbf{x}, R) = \frac{1}{2} \cdot \arg\max_{R' \geq R}\{\mathcal{L}(f, \mathcal{N}(\mathbf{x}, R')) \leq \Delta_{LS}^p\}$$

If there is no solution to the equation above, then we return 0.

**Lemma 4.2.** $\phi(\mathbf{x}, R) \leq \gamma(\mathbf{x}, R)$. *Proof in Appendix B.2.*

**Privately testing the lower bound**: The next step in the PTR algorithm requires testing if $\gamma(\mathbf{x}) \leq \ln(\frac{1}{\delta})/\epsilon$. If the condition is true, then no-answer ($\perp$) is released instead of data. Since the $\gamma$ query depends upon $\mathbf{x}$, PTR privatizes it by applying laplace mechanism, i.e. $\hat{\gamma}(\mathbf{x}) = \gamma(\mathbf{x}) + \mathsf{Lap}(1/\epsilon)$. The query has a sensitivity of 1 since the $\gamma$ could differ at most by 1 for any two neighboring databases. In our framework, we compute $\phi(\mathbf{x}, R)$ to lower bound the value of $\gamma(\mathbf{x}, R)$. Therefore, we need to privatize the $\phi$ query. For general distance metrics in $d_\chi$-privacy, the global sensitivity of the $\phi(\mathbf{x})$ query is 1.

**Lemma 4.3.** *The query $\phi(\cdot)$ has a global sensitivity of 1, i.e. $\forall \mathbf{x}, \mathbf{x}' \in \chi, d_{abs}(\phi(\mathbf{x}, R), \phi(\mathbf{x}', R)) \leq d_\chi(\mathbf{x}, \mathbf{x}')$. Proof in Appendix B.3.*

After computing $\phi(\mathbf{x}, R)$, we add noise sampled from a laplace distribution, i.e. $\hat{\phi}(\mathbf{x}, R) = \phi(\mathbf{x}, R) + \mathsf{Lap}(R/\epsilon)$. Next, we check if $\hat{\phi}(\mathbf{x}, R) \leq \ln(\frac{1}{\delta}) \cdot R/\epsilon$, then we release $\perp$, otherwise we release $\hat{\mathbf{z}} = f(g(\mathbf{x})) + \mathsf{Lap}(\Delta_{LS}^p/\epsilon)$. Next, we prove that the mechanism $\mathcal{M}_1$ described above satisfies *semantic neighborhood-privacy*.

**Theorem 4.4.** *Mechanism $\mathcal{M}_1$ satisfies uniform $(2\epsilon, \delta/2, R)$-semantic neighborhood privacy Eq. 3, i.e. $\forall \mathbf{x}, \mathbf{x}' \in \chi, s.t. d_\chi(\mathbf{x}, \mathbf{x}') \leq R$*

$$\mathbb{P}(\mathcal{M}(\mathbf{x}) \in S) \leq e^{2\epsilon}\mathbb{P}(\mathcal{M}(\mathbf{x}') \in S) + \frac{\delta}{2} \tag{9}$$

Proof Sketch: Our proof is similar to the proof for the PTR framework(Dwork et al. (2014)) except the peculiarity introduced due to our metric space formulation. First, we show that not releasing the answer ($\perp$) satisfies the privacy definition. Next, we divide the proof into two parts, when the proposed bound is incorrect (i.e. $\Delta_{LS}(\mathbf{x}, R) > \Delta_{LS}^p$) and when it is correct. Let $\hat{R}$ be the output of query $\phi$.

$$\frac{\mathbb{P}[\hat{\phi}(\mathbf{x}, R) = \hat{R}]}{\mathbb{P}[\hat{\phi}(\mathbf{x}', R) = \hat{R}]} = \frac{\exp(-(\frac{|\phi(\mathbf{x}, R) - \hat{R}|}{R} \cdot \epsilon))}{\exp(-(\frac{|\phi(\mathbf{x}', R) - \hat{R}|}{R} \cdot \epsilon))} \leq \exp(|\phi(\mathbf{x}', R) - \phi(\mathbf{x}, R)| \cdot \frac{\epsilon}{R}) \leq \exp(d_\chi(\mathbf{x}, \mathbf{x}') \cdot \frac{\epsilon}{R})$$

$$\leq \exp(\epsilon)$$

Therefore, using the post-processing property - $\mathbb{P}[\mathcal{M}(\mathbf{x}) = \perp] \leq e^\epsilon \mathbb{P}[\mathcal{M}(\mathbf{x}') = \perp]$. Here, the first inequality is due to triangle inequality, the second one is due to Lemma 4.3 and the third inequality follows from $d_\chi(\mathbf{x}, \mathbf{x}') \leq R$. Note that when $\Delta_{LS}(\mathbf{x}, R) > \Delta_{LS}^p$, $\phi(\mathbf{x}, R) = 0$. Therefore, the probability for the test to release the answer in this case is

$$\mathbb{P}[\mathcal{M}(\mathbf{x}) \neq \perp] = \mathbb{P}[\phi(\mathbf{x}, R) + \mathsf{Lap}(\frac{R}{\epsilon}) > \log(\frac{1}{\delta}) \cdot \frac{R}{\epsilon}] = \mathbb{P}[\mathsf{Lap}(\frac{R}{\epsilon}) > \log(\frac{1}{\delta}) \cdot \frac{R}{\epsilon}]$$

Based on the CDF of Laplace distribution, $\mathbb{P}[\mathcal{M}(\mathbf{x}) \neq \perp] = \frac{\delta}{2}$. Therefore, if $\Delta_{LS}(\mathbf{x}, R) > \Delta_{LS}^p$, for any $S \subseteq \mathbb{R}^d \cup \perp$ in the output space of $\mathcal{M}$

$$\mathbb{P}[\mathcal{M}(\mathbf{x}) \in S] = \mathbb{P}[\mathcal{M}(\mathbf{x}) \in S \cap \{\perp\}] + \mathbb{P}[\mathcal{M}(\mathbf{x}) \in S \cap \{\mathbb{R}^d\}]$$

| | MNIST ($\epsilon = 0, 0.10$), ($\epsilon = \infty, 0.93$) | | | | | FMNIST ($\epsilon = 0, 0.10$), ($\epsilon = \infty, 0.781$) | | | | | UTKFace ($\epsilon = 0, 0.502$), ($\epsilon = \infty, 0.732$) | | | | |
|---|---|---|---|---|---|---|---|---|---|---|---|---|---|---|---|
| | Informal | $\epsilon=1$ | $\epsilon=2$ | $\epsilon=5$ | $\epsilon=10$ | Informal | $\epsilon=1$ | $\epsilon=2$ | $\epsilon=5$ | $\epsilon=10$ | Informal | $\epsilon=1$ | $\epsilon=2$ | $\epsilon=5$ | $\epsilon=10$ |
| Encoder | **0.93** | 0.428 | 0.673 | 0.883 | **0.921** | 0.779 | 0.228 | 0.355 | 0.605 | **0.722** | 0.724 | 0.617 | 0.673 | 0.717 | 0.721 |
| ARL | 0.917 | 0.329 | 0.532 | 0.792 | 0.882 | 0.747 | 0.214 | 0.319 | 0.557 | 0.685 | 0.71 | 0.605 | 0.649 | 0.691 | 0.707 |
| C | 0.926 | 0.443 | 0.684 | 0.881 | 0.917 | **0.781** | 0.158 | 0.225 | 0.422 | 0.608 | **0.73** | 0.623 | 0.673 | **0.718** | **0.724** |
| N | 0.923 | 0.279 | 0.496 | 0.816 | 0.902 | 0.559 | 0.136 | 0.177 | 0.310 | 0.462 | 0.725 | 0.614 | 0.667 | 0.708 | 0.715 |
| ARL-C | 0.896 | 0.424 | 0.648 | 0.839 | 0.883 | 0.761 | 0.196 | 0.314 | 0.537 | 0.682 | 0.709 | 0.632 | 0.684 | 0.70 | 0.705 |
| ARL-N | 0.88 | 0.118 | 0.139 | 0.21 | 0.325 | 0.717 | 0.294 | 0.467 | 0.657 | 0.705 | 0.71 | 0.628 | 0.674 | 0.701 | 0.708 |
| C-N | 0.929 | 0.353 | 0.574 | 0.844 | 0.913 | 0.774 | 0.161 | 0.224 | 0.411 | 0.599 | 0.727 | 0.616 | 0.671 | 0.712 | 0.722 |
| ARL-C-N | 0.921 | **0.514** | **0.751** | **0.891** | 0.912 | 0.706 | **0.371** | **0.554** | **0.678** | 0.695 | 0.712 | **0.650** | **0.690** | 0.700 | 0.700 |

Table 1: **Performance comparison for different baselines:** Our posthoc framework enables comparison between different obfuscation techniques by fixing the privacy budget ($\epsilon$). First four rows are different approaches to protect against data reconstruction and the remaining rows below are combinations of different approaches. The top row refers to the accuracy corresponding to different datasets under two extremes of epsilons. ARL refers to widely used adversarial representation learning approach for regularizing representation based on a proxy attacker(Li et al. (2021); Liu et al. (2019); Xiao et al. (2020); Singh et al. (2021)). Contrastive refers to contrastive learning based informally privatizing mechanism introduced in Osia et al. (2020).

$$\leq e^{\epsilon}\mathbb{P}[\mathcal{M}(\mathbf{x}') \in S \cap \{\bot\}] + \mathbb{P}[\mathcal{M}(\mathbf{x}) \neq \bot] \leq e^{\epsilon}\mathbb{P}[\mathcal{M}(\mathbf{x}') \in S] + \frac{\delta}{2}$$

If $\Delta_{LS}(\mathbf{x}, R) \leq \Delta_{LS}^{p}$ then the mechanism is a composition of two $(\epsilon, \delta, R)$-semantic neighborhood private algorithm where the first algorithm ($\phi(\mathbf{x}, R)$) is $(\epsilon, \delta/2, R)$-semantic neighborhood private and the second algorithm is $(\epsilon, 0, R)$-private. Using composition, the algorithm is $(2\epsilon, \delta/2, R)$-semantic neighborhood private. We describe $\mathcal{M}_1$ step by step in Algorithm 1. To summarize, we designed the posthoc privacy framework that extends the PTR framework by making it tractable to get $(\epsilon, \delta, R)$-semantic neighborhood privacy. The exact local Lipschitz constant of the neural network based obfuscator is estimated using mixed-integer programming based optimization developed by Jordan & Dimakis (2020).

**Computational feasibility**: Our key idea is to add extra computation on the client side to formally reason about the privacy of shared data. This extra computational cost is due to the estimation of the local Lipschitz constant of the *obfuscator* network. However, three key factors of our framework make it practically feasible -

1. We compute the local Lipschitz constant (i.e. in a small neighborhood around a given point): Our extension of the propose-test-release framework only requires us to operate in a small local neighborhood instead of estimating the global Lipschitz constant which would be much more computationally expensive.

2. Low number of parameters for obfuscator: Instead of estimating the Lipschitz constant of the whole prediction model, we only require estimation of the obfuscator - a neural network that has a significantly lower number of parameters in comparison to the prediction model.

3. Lower dimension of input embedding: Since we measure distance in the embedding space, the dimensions over which the local Lipschitz constant is estimated are significantly lower than the ambient data dimension.

We performed an ablation study on all three aspects mentioned above in Sec 6. The fact that the local Lipschitz constant is being computed over the same obfuscator allows room for optimizing performance by caching. Our goal is to demonstrate the feasibility of bridging formal privacy guarantees and ARL-based mechanisms, hence, we did not explore such performance speedups.

## 5 EXPERIMENTS

**Experimental Setup:** We evaluate different aspects of our proposed framework - i) **E1**: comparison between different adversarial appraches, ii) **E2**: comparison with local differential privacy (LDP), iii) **E3**: computational tractability of our proposed framework, and iv) **E4**: investigating role of ARL in achieving privacy. We use MNIST(LeCun (1998)), FMNIST(Xiao et al. (2017))

and UTKFace(Zhang et al. (2017)) dataset for all experiments. All of them contain samples with extremely high ambient dimensions (MNIST-784, FMNIST-784 and UTKFace-4096). We use a deep CNN based $\beta$-VAE(Higgins et al. (2016)) for the *embedder*. We use LipMip(Jordan & Dimakis (2020)) for computing Lipschitz constant over $\ell_\infty$ norm in the input space and $\ell_1$ norm in the output space. We baseline with a simple *Encoder* based approach where the data is projected to smaller dimensions using a neural network. This encoder type approach has been used in the literature as Split Learning Gupta & Raskar (2018). For ARL, we use the proxy-adversary based min-max optimization used by several ARL techniques(Xiao et al. (2020); Singh et al. (2021); Liu et al. (2019); Li et al. (2021)) and adversarial contrastive learning(Osia et al. (2020)) which we denote as C. We use noisy regularization (denoted by N) to improve classifier performance. We refer the reader to sec E for a detailed experimental setup, codebase and hyper-parameters.

**E1: Privacy-Utility Trade-off:** Since our framework enables comparison between different obfuscation techniques under same privacy budget, we evaluate test set accuracy on three image datasets in Table 1. Our results indicate that ARL complemented with contrastive and noise regularization helps in attaining overall best performance among all possible combinations. We note that the SoTA performance on all three datasets is higher than our experimental setup because of the usage of *embedder* that can be further improved to yield higher accuracy.

**E2: Comparison between ARL and LDP:** While $\epsilon$-LDP definition provides a different and stronger privacy guarantee than our proposed privacy definition, we compare the performance between ARL and LDP and report results in the Table 2 in the Appendix for the sake of completeness. Results indicate that for low value of $\epsilon$, LDP techniques do not yield any practical utility. This observation corroborates with impossibility result of instance encoding Carlini et al. (2020) and our discussion in Sec 2 about applicability of traditional DP in the context of ARL.

**E3: Computational feasibility:** Our framework relies upon the exact computation of Lipschitz constant of ReLU networks (obfuscator in our case) that has been shown to be a NP-hard problem(Jordan & Dimakis (2020)). Our end-to-end runtime evaluation on a CPU based client results in a runtime of **2 sec/image** (MNIST) and **3.5 sec/image** (UTKFace). While plenty of room exists for optimizing this runtime, we believe current numbers serve as a reasonable starting point for providing formal privacy in ARL. As discussed in Sec 4, we compare computation time of the *obfuscator* across three factors relevant to our setup - i) Dimensionality of the input, ii) Size of the neighborhood, and iii) Number of layers in the *Obfuscator*. Figure 4 shows performance evaluation. While the running time quickly grows exponentially with input size, we emphasize that the *obfuscator* network requires only small number of dimensions due to its input residing on the embedding space. Results demonstrate that not only the framework is computationally tractable but it can be executed at a real-time speed for our inference use-case.

**E4: What role does ARL play in achieving privacy?** In this experiment, we assess the contribution of adversarial training in improving privacy-utility trade-off. We train *obfuscator* models with different values of $\alpha$ (weighing factor) for adversarial training. Our results in Fig 2 indicate that higher weighing of adversarial regularization reduces the local lipschitz constant, hence reducing the local sensitivity of the neural network. Furthermore, for high values of $\alpha$, the change in local lipschitz constant reduces significantly for different size ($R$) of the neighborhood. These two observations can potentially explain that ARL improves reconstruction privacy by reducing the sensitivity of the *obfuscator*. However, as we observe in Table 1, the classifier can reduce its utility if ARL is not complemented with noisy and contrastive regularization. We believe this finding could be of independent interest for the adversarial defense community where the goal is to reduce misclassification performance of neural networks.

## 6 DISCUSSION

**How to select privacy parameter $R$?** One of the key difference between $(\epsilon, \delta, R)$-neighborhood privacy and $(\epsilon, \delta)$-DP is the additional parameter $R$. The choice of $R$ depends upon the neighborhood in which a user wishes to get an $\epsilon$ level of indistinguishability. We perform reconstruction attacks on privatized encoding obtained from our framework by training an ML model to reconstruct original images. We compare reconstruction results for different values of $\epsilon$ and $R$ on four distinct metrics for images in Table 4,5,3.6. To assess the level of indistinguishability, we look at Fig 3 where we project the original images into embedding space and sample points from the boundary of

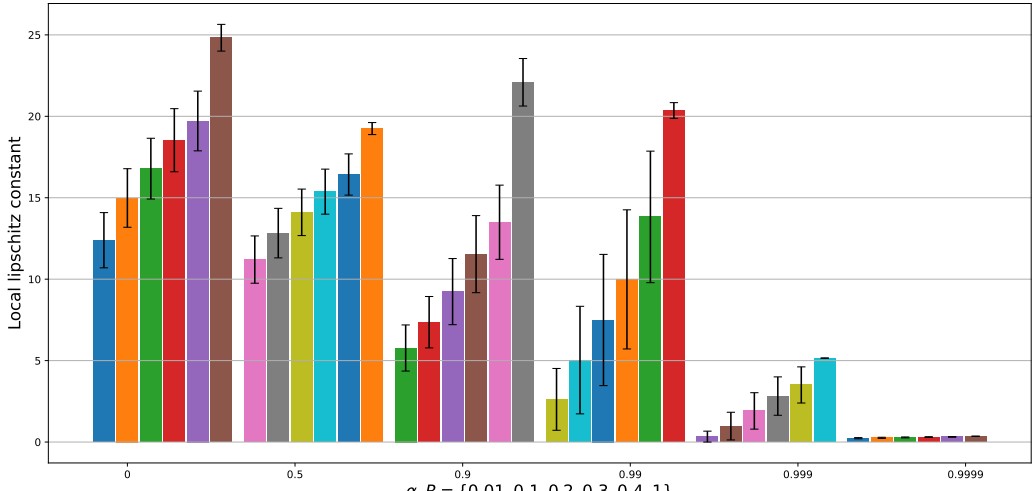

Figure 2: **Local sensitivity comparison for different values of** $\alpha$: The five bars for each $\alpha$ represent different neighborhood radii. Increase in the value of $\alpha$ decreases the local Lipschitz constant (upper bound on local sensitivity) indicating lesser amount of noise to be added for the same level of privacy.

neighborhoods of different $R$. We observe that as the boundary of the neighborhood increases, the images become perceptually different from the original image. For extremely large radii, the images change significantly enough that their corresponding label may change too. Such visualization can be used to semantically understand different values of $R$.

**How to propose $\Delta_{LS}^p$?** Our framework requires a proposal on the upper bound of local sensitivity in a private manner. One possible way to obtain $\Delta_{LS}^p$ is by using the Lipschitz constant of training data samples used in training the *obfuscator*. To incorporate this notion of average, we choose $\Delta_{LS}^p$ by first computing the mean ($\mu$) and standard deviation ($\sigma$) of local sensitivity on the training dataset (assumed to be known under our threat model), then we keep $\Delta_{LS}^p = \mu + n * \sigma$ where $n$ allows a trade-off between the likelihood of releasing the samples under PTR and adding extra noise to data. We used $n = 3$ in our experiments. Since empirically, the value of local sensitivity appears to be following a gaussian, using confidence interval serves as a good proxy. Fig 2 shows that for higher values of $\alpha$, the variability in the local Lipschitz constant decreases indicating the validity of the bound would hold for a large number of samples. We emphasize that privacy parameters should be chosen independently of the private data otherwise the guarantees do not hold.

**Limitations:** i) The distance metric ($d_\theta^\beta(\mathbf{x}, \mathbf{x}')$) is currently learned from data and could lead to irrelevant privacy guarantees if semantically similar points are farther apart in the embedding space. We believe this limitation could be addressed by understanding the convergence of these learned distance metrics. Furthermore, these learned distance metrics might be better than not assessing privacy formally at all or using distance metrics like $\ell_1, \ell_2$ norm in the ambient dimension of data. ii) Since we utilize the PTR framework, outlier samples may not get released due to high sensitivity, this is expected since these outlier samples are likely to be misclassified anyway. iii) Lipschitz constant computation is limited to ReLU networks, therefore more sophisticated obfuscator architectures are currently not compatible with our proposed framework.

## 7    CONCLUSION

ML based approaches to private inference have been on a rise in the past few years owing to the powerful representational capacity of neural networks, especially for complex real-world datasets. Their main drawback is the lack of formal privacy guarantees. Our work has taken the first steps towards a formal privacy guarantee for a broad class of existing empirical techniques for privacy. We believe that our framework would foster more research in ARL techniques by improving privacy-utility evaluation and take them closer to real world adoption.

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

# A    RELATED WORK

**ARL** techniques aim to *learn* a task-oriented privacy preserving encoding of data. Majority of the works in this area either protect against sensitive attribute leakage Hamm (2017); Roy & Boddeti (2019); Bertran et al. (2019); Li et al. (2018) or input reconstruction Samragh et al. (2021); Singh et al. (2021); Mireshghallah et al. (2021); Li et al. (2021); Liu et al. (2019). These techniques usually evaluate their privacy using empirical attacks since the mechanism is learned using gradient based min-max optimization making it infeasible for the worst-case privacy analysis. The goal of our work is to make them amenable to formal privacy analysis. While theoretical analyses Zhao et al. (2020a;b); Sadeghi & Boddeti (2021) of ARL objectives have identified fundamental trade-offs between utility and attribute leakage, they are difficult to formalize as a worst-case privacy guarantee especially for deep neural networks.

**Privacy definitions** that extend the DP definition to incorporate some of its limitations Kifer & Machanavajjhala (2011) include $d_\chi$-Privacy Chatzikokolakis et al. (2013), and Pufferfish Kifer & Machanavajjhala (2014).   Our privacy definition is a specific instantiation of the $d_\chi$-Privacy Chatzikokolakis et al. (2013) framework that extends DP to general metric spaces. Our instantiation is focused on reconstruction privacy for individual samples instead of membership inference attacks Dwork et al. (2017). Existing works in DP for reconstruction attacks Bhowmick et al. (2018); Stock et al. (2022) focus on the privacy of training data.

**Lipschitz constant** estimation for neural networks has been used to guarantee network's stability to perturbations. Existing works either provide an upper bound Weng et al. (2018); Latorre et al. (2020); Fazlyab et al. (2019), exact Lipschitz constant Jordan & Dimakis (2020; 2021) or Lipschitz constant regularization Scaman & Virmaux (2018); Huang et al. (2021) during the training stage. Some existing works have explored the relationship between adversarial robustness and DP model training Phan et al. (2020); Pinot et al. (2019); Tursynbek et al. (2020). We utilize similar ideas of perturbation stability but for privacy. Shavit and Gjura Shavit & Gjura (2019) use Lipschitz neural networks Gouk et al. (2018) to learn a private mechanism design for summary statistics such as median, however their mechanism design lack privacy guarantee.

**Posthoc approach to privacy** applies privacy preserving mechanism in a data dependent manner. Smooth sensitivity Nissim et al. (2007) and PTR Dwork & Lei (2009) reduce the noise magnitude since local sensitivity is only equal to global sensitivity in the worst case. Privacy odometer Rogers et al. (2016), Ex-post privacy loss Ligett et al. (2017) and Rényi privacy filter Feldman & Zrnic (2021) track privacy loss as the query is applied over data. Our works builds upon the PTR framework in order to give high privacy for less sensitive data. However, as we show in Sec 4, our framework reformulates the PTR algorithm to make it tractable under our setup.

# B    PROOFS

**Lemma B.1.** *For a given $f$ and for $d_\mathcal{Y} \leftarrow \ell_\alpha$ and $d_\chi \leftarrow \ell_\beta$, $\Delta_{LS}(\mathbf{x}, R) \leq \mathcal{L}(f, \mathcal{N}(\mathbf{x}, R))$.*

*Proof.* Local sensitivity ($\Delta_{LS}$) for a sample $\mathbf{x}$ in a radius $R$ for a query $f$ is defined as:

$$\Delta_{LS}(\mathbf{x}, R) = \sup_{\mathbf{x}' \in \mathcal{N}(\mathbf{x}, R)} \frac{d_\mathcal{Y}(f(\mathbf{x}), f(\mathbf{x}'))}{d_\chi(\mathbf{x}, \mathbf{x}')}$$

Local Lipschitz constant ($\mathcal{L}$) for a function $f$ and a neighborhood $\mathcal{N}$ is defined as:

$$\mathcal{L}^{\alpha,\beta}(f, \mathcal{N}) = \sup_{\mathbf{x}', \mathbf{x}'' \in \mathcal{N}} \frac{||f(\mathbf{x}') - f(\mathbf{x}'')||_\alpha}{||\mathbf{x}' - \mathbf{x}''||_\beta} \quad (\mathbf{x}' \neq \mathbf{x}'')$$

If $\mathcal{L}$ is defined around neighborhood $\mathcal{N}(\mathbf{x}, R)$ then the set over which local sensitivity is computed is a subset of the set over which local Lipschitz constant is estimated. Intuitively, local Lipschitz condition is for all possible pair of samples in the neighborhood while local sensitivity is for all samples with respect to the given sample. Since both conditions require a suprememum over the set, $\Delta_{LS}(\mathbf{x}, R) \leq \mathcal{L}(f, \mathcal{N}(\mathbf{x}, R))$. □

**Lemma B.2.** *Algorithm $\phi$ gives a lower bound on the query $\gamma$. That is, $\phi(\mathbf{x}, R) \leq \gamma(\mathbf{x}, R)$.*

*Proof.* The $\gamma$ query is defined as -

$$\gamma(\mathbf{x}, R) = \min_{\mathbf{x}' \in \chi} \{ d_\chi(\mathbf{x}, \mathbf{x}') \ \ s.t. \ \ \Delta_{LS}(\mathbf{x}', R) > \Delta_{LS}^p \}. \tag{10}$$

The $\phi$ query is defined as -

$$\phi(\mathbf{x}, R) = \frac{1}{2} \cdot \arg\max_{R' \geq R} \{ \mathcal{L}(f, \mathcal{N}(\mathbf{x}, R')) \leq \Delta_{LS}^p \} \tag{11}$$

For any given sample $\mathbf{x}$ and privacy parameters $(R, \Delta_{LS}^p)$ such that $s = \phi(\mathbf{x}, R)$, we know that $\forall \mathbf{x}' \in \mathcal{N}(\mathbf{x}, s)$

$$\mathcal{N}(\mathbf{x}', s) \subset \mathcal{N}(\mathbf{x}, 2s)$$
$$\implies \mathcal{L}(f, \mathcal{N}(\mathbf{x}', s)) \leq \mathcal{L}(f, \mathcal{N}(\mathbf{x}, 2s))$$

Based on eq 10, we know that $\mathcal{L}(f, \mathcal{N}(\mathbf{x}, 2s)) \leq \Delta_{LS}^p$ and hence $\forall \mathbf{x}' \in \mathcal{N}(\mathbf{x}, s)$,

$$\mathcal{L}(f, \mathcal{N}(\mathbf{x}', s)) \leq \Delta_{LS}^p$$

Therefore, $\Delta_{LS}(\mathbf{x}', R) \leq \Delta_{LS}^p$ and hence,

$$s \leq \gamma(\mathbf{x})$$

For the cases when there is not any feasible solution, $\phi$ returns 0 which is exactly the same answer for $\gamma$ query. This completes the proof. $\qquad\square$

**Lemma B.3.** *The query $\phi(\cdot)$ has a global sensitivity of* 1, *i.e.* $d_{abs}(\phi(\mathbf{x}, R), \phi(\mathbf{x}', R)) \leq d_\chi(\mathbf{x}, \mathbf{x}')$

*Proof.* We will prove the above argument through a contradiction. We will prove that for a fixed radius $R$ and any arbitrary point $\mathbf{x} \in \chi$, the neighborhood spanned by $\phi(\mathbf{x}, R)$ can not be a proper superset for any neighborhood spanned by any other point $\phi(\mathbf{x}', R)$. More formally, we will prove, $\forall \mathbf{x}, \mathbf{x}' \in \chi$, $\overline{\mathcal{N}(\mathbf{x}, \phi(\mathbf{x}, R))} \not\subset \mathcal{N}(\mathbf{x}', \phi(\mathbf{x}', R))$ and $\mathcal{N}(\mathbf{x}', \phi(\mathbf{x}', R)) \not\subset \mathcal{N}(\mathbf{x}, \phi(\mathbf{x}, R))$. Once proven, this argument allows us to specify the distance between $\mathbf{x}$ and $\mathbf{x}'$ with respect to $\phi(\mathbf{x}, R)$ and $\phi(\mathbf{x}', R)$. Since the function $\phi(\mathbf{x}, R)$ returns the maximum possible value such that

$$\mathcal{L}(f, \mathcal{N}(\mathbf{x}, \phi(\mathbf{x}, R))) \leq \Delta_{LS}^p$$

Therefore, for any $\zeta > 0$

$$\mathcal{L}(f, \mathcal{N}(\mathbf{x}, \phi(\mathbf{x}, R) + \zeta)) > \Delta_{LS}^p \tag{12}$$

For contradiction, we assume that $\exists \mathbf{x}, \mathbf{x}' \in \chi$ s.t. $\mathcal{N}(\mathbf{x}, \phi(\mathbf{x}, R)) \subset \mathcal{N}(\mathbf{x}', \phi(\mathbf{x}', R))$

$$\implies \exists \, \eta > 0 \ s.t. \ \mathcal{N}(\mathbf{x}, \phi(\mathbf{x}, R) + \eta) \subseteq \mathcal{N}(\mathbf{x}', \phi(\mathbf{x}', R)) \tag{13}$$

$$\implies \mathcal{L}(\mathcal{N}(\mathbf{x}, \phi(\mathbf{x}, R) + \eta)) \leq \Delta_{LS}^p \tag{14}$$

This leads to a contradiction between eq 12 and eq 14. Therefore, $\forall \mathbf{x}, \mathbf{x}' \in \chi$,

$$\phi(\mathbf{x}, R) \leq \phi(\mathbf{x}', R) + d_\chi(\mathbf{x}, \mathbf{x}')$$

Using symmetry argument, we can show that

$$d_{abs}(\phi(\mathbf{x}, R), \phi(\mathbf{x}', R)) \leq d_\chi(\mathbf{x}, \mathbf{x}')$$

This completes the proof. $\qquad\square$

## C  PROPOSE-TEST-RELEASE

DP mechanisms typically add noise based on the global sensitivity of a query. However, for several queries over various data distributions, average local sensitivity might be much lower than the global sensitivity. However, local sensitivity is data dependent hence the amount of noise introduced by a mechanism based on local sensitivity itself can reveal private information. Therefore, to add noise based on local sensitivity in a privacy preserving manner, Dwork & Lei (2009) introduced PTR. Conceptually, the idea behind PTR is to propose an arbitrary upper bound on the true value of local sensitivity. This upper bound should be obtained privately otherwise the choice of upper bound itself can reveal private information. To test whether the proposed bound is correct, the mechanism performs a privacy-preserving testing of the upper bound. The test itself is a randomized algorithm due to privacy requirements. Therefore, it can have false positives and false negatives. If the test fails, the mechanism returns $\perp$ (no-answer). Otherwise, standard DP mechanism (ex. - Laplace) is applied to the query based on the proposed sensitivity (and not true local sensitivity). More formally, the algorithm proceeds in the following steps -

1. A proposal on upper bound of a query $q$ is fed as input for data $x$. Let us call it $\Delta_{LS}^p$.

2. The algorithm finds the closest point $x'$ to $x$ such that $\Delta_{LS}(x') > \Delta_{LS}^p$. Here $\Delta_{LS}$ refers to local sensitivity for the query $q$.

3. Let $\gamma = d_{\mathcal{H}}(x, x')$ and $\hat{\gamma} = \gamma + Lap(1/\epsilon)$

4. If $\hat{\gamma} \leq ln(1/\delta)/\epsilon$; return $\bot$

5. Else share data $x + Lapl(\Delta_{LS}^p/\epsilon)$

**Computational Cost:** Depending on the query, this algorithm can incur significant computation cost. Especially in the Step 2, finding closest $x'$ can be impractical if the data space is high dimensional. Finally, for queries such as neural networks evaluating local sensitivity itself is not practical since it requires giving exact and correct solution to a non-convex optimization problem. Therefore, our framework relies on computing local lipschitz constant over a small neighborhood instead of local sensitvity over the complete data space.

---

**Algorithm 1:** Extended PTR algorithm for $(\epsilon, \delta, R)$-semantic neighborhood privacy

**Data:** $\mathbf{x} \in \mathcal{X}$
**Inputs:** $\epsilon \in \mathbb{R}^+, \delta \in \mathbb{R}^+, R \in \mathbb{R}^+, \Delta_{LS}^p \in \mathbb{R}^+$
**Init:** $\zeta \in \mathbb{R}^+$ ;      /* For numerical stability, typically very small */
**Init:** $R_{min} = R, R_{max} \in \mathbb{R}^+$
**while** $R_{max} > R_{min} + \zeta$ **do**
    $R_{mid} = (R_{min} + R_{max})/2$;
    $r = \mathcal{L}(f, \mathcal{N}(\mathbf{x}, R))$ ;          /* Compute local Lipschitz constant */
    **if** $r < \Delta_{LS}^p$ **then** $R_{min} = R_{mid}$; **else** $R_{max} = R_{mid}$; **end**
**end**
$\hat{r} = \frac{R_{min}}{2}$;
$\hat{R} \leftarrow \hat{r} + \mathsf{Lap}(1/\epsilon)$;
**if** $\hat{R} < ln(1/\delta)/\epsilon$ **then** return $\bot$; **else** return $f(\mathbf{z}) + \mathsf{Lap}(\Delta_{LS}^p/\epsilon)$; **end**

---

## D    LIPSCHITZ CONSTANT ESTIMATION

We use the mixed integer programming based algorithm LipMip by Jordan & Dimakis (2020) for computing the local Lipschitz constant. Their technique allows exactly computing the local Lipschitz constant of a neural network with ReLU non-linearities. Their key idea is to estimate the supremal norm of the jacobian of the neural network. Since ReLU networks do not allow for differentiability, LipMip uses clark jacobian to circumvent the issue and encode the optimization objective of obtaining local Lipschitz constant over a pre-defined neighborhood as a mixed integer programming problem. The neighborhood is specified as a hypercube with same dimension as points in the neighborhood. Their algorithm searches for feasible regions and minimizes the gap between lower and upper bound on the Lipschitz constant.

## E    EXPERIMENTAL DETAILS

Our experimental setup operates in three stages - i) Embedder training, ii) Obfuscator training, and iii) Private inference. Our codebase is available `https://drive.google.com/drive/folders/1DpHhS9u-Mpp3TVmTYiue7BKKUshyKw2w?usp=sharing` here for reproducability. We will release the code and all trained models publicly after the reviews. For all our experiments we use PyTorch(Paszke et al. (2019)) with Nvidia-GeForce GTX TITAN GPU. We use $\beta$-VAE with $\beta = 5$ for the design of the *embedder*.

1. **Embedder Training:** We use embedding dimension as 8 for MNIST and FMNIST dataset. For the UTKFace dataset, we use embedding size as 10. We use Adam optimizer(Kingma & Ba (2014)) with a constant learning rate of 0.001. The VAE architecture for MNIST and

|  |  | $\epsilon = 1$ | $\epsilon = 2$ | $\epsilon = 5$ | $\epsilon = 7$ | $\epsilon = \infty$ |
|---|---|---|---|---|---|---|
| MNIST | LDP-Image | 0.1 | 0.1 | 0.1 | 0.3 | 0.99 |
|  | LDP-Embedding | 0.1075 | 0.1319 | 0.1927 | 0.3107 | 0.9096 |
|  | Adversarial | 0.514 | 0.751 | 0.891 | 0.912 | 0.9291 |
| FMNIST | LDP-Image | 0.1 | 0.1 | 0.1 | 0.1 | 0.92 |
|  | LDP-Embedding | 0.1012 | 0.1251 | 0.1708 | 0.2420 | 0.7798 |
|  | Adversarial | 0.371 | 0.554 | 0.678 | 0.722 | 0.781 |
| UTKFace | LDP-Image | 0.52 | 0.52 | 0.52 | 0.52 | 0.89 |
|  | LDP-Embedding | 0.5040 | 0.535 | 0.5757 | 0.6375 | 0.7246 |
|  | Adversarial | 0.65 | 0.69 | 0.718 | 0.724 | 0.73 |

Table 2: Comparison between LDP and Adversarial Representation Learning: Using our proposed framework we compare the utility of LDP and ARL across different values of the privacy parameter $\epsilon$.

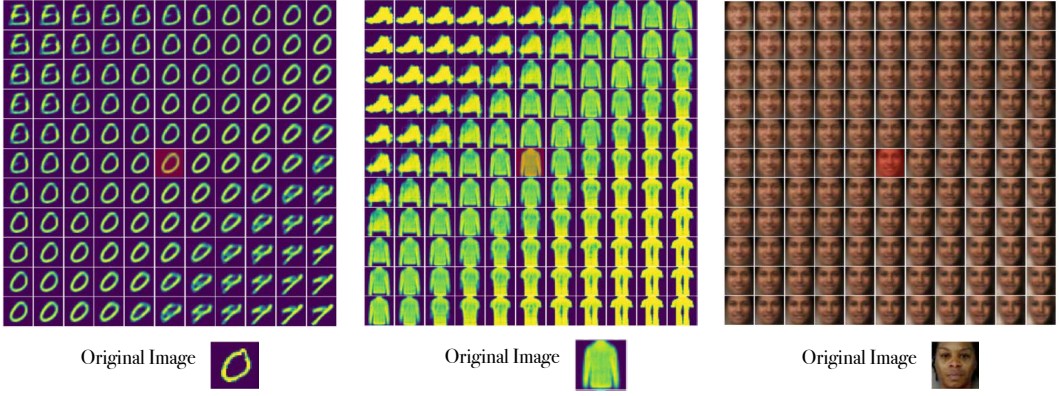

Figure 3: Neighborhood for different image datasets. The center image (in translucent red) is the reconstruction of the original image with nearby images sampled from the embedding space. Note that there are multiple dimensions and we have illustrated interpolation for only one here.

FMNIST dataset is composed of three fully connected layers with non-linear activations and dropout.

2. **Obfuscator Training:** For ARL, we use $\alpha = 0.99$, for noisy regularization, we use $\sigma = 0.01$ and for contrastive regularization, we use $\lambda = 1.0$ with a *margin* of 25. All of these regularizations are trained jointly using Adam optimizer(Kingma & Ba (2014)).

3. **Private Inference:** In the this stage we use LipMip(Jordan & Dimakis (2020)) which is built upon Gurobi Optimizer(Gurobi Optimization, LLC (2022)) for solving the Mixed-Integer programming formulation of local lipschitz constant estimation. For the metrics, we use $d_\mathcal{X}$ as infinity norm and $d_\mathcal{Y}$ as $\ell_1$-norm. For the privacy parameters, we use $\delta = 0.05$ and $R = 0.5$ for MNIST, $R = 0.2$ for FMNIST and $R = 0.1$ for UTKFace. The choice of different $R$ was based on visualizing samples from the training set and evaluating how far similar looking samples lie in the embedding space.

| SSIM | MNIST |  |  |  |  |  | UTKFace |  |  |  |  |  |
|---|---|---|---|---|---|---|---|---|---|---|---|---|
|  | 0.1 | 0.2 | 0.4 | 0.6 | 0.8 | 1.0 | 0.1 | 0.2 | 0.4 | 0.6 | 0.8 | 1.0 |
| $\epsilon=1$ | 0.558 | 0.427 | 0.287 | 0.231 | 0.201 | 0.179 | 0.481 | 0.443 | 0.426 | 0.425 | 0.422 | 0.421 |
| $\epsilon=2$ | 0.648 | 0.553 | 0.414 | 0.328 | 0.275 | 0.229 | 0.507 | 0.475 | 0.442 | 0.439 | 0.428 | 0.426 |
| $\epsilon=5$ | 0.702 | 0.655 | 0.580 | 0.499 | 0.434 | 0.380 | 0.519 | 0.5037 | 0.481 | 0.465 | 0.451 | 0.447 |
| $\epsilon=10$ | 0.710 | 0.700 | 0.656 | 0.610 | 0.560 | 0.517 | 0.519 | 0.5169 | 0.507 | 0.494 | 0.485 | 0.476 |

Table 3: SSIM metric for reconstruction attack with varying $R$ and $\epsilon$

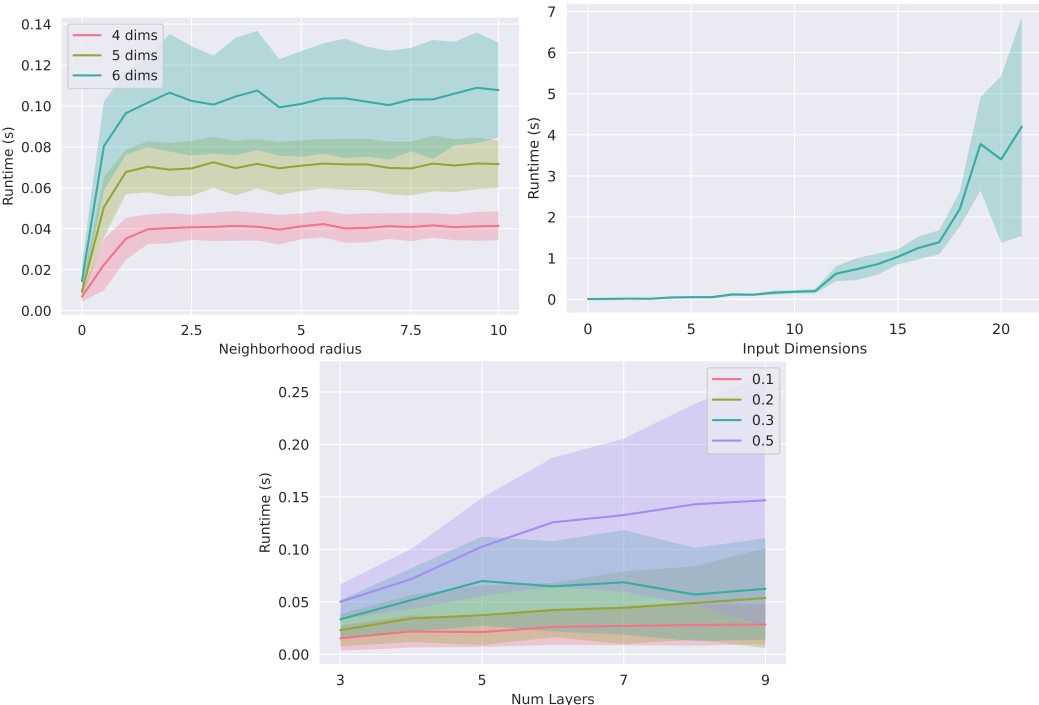

Figure 4: **Runtime evaluation of local lipschitz computation** for different (a) neighborhood radius, (b) input dimensions, and (c) number of layers. While the runtime increases exponentially with dimensions, it plateaus with increase in neighborhood radius. Since the input dimensions are same as embedding dimensions making the algorithm favorable to our analysis.

| $\ell_1$ | MNIST | | | | | | UTKFace | | | | | |
|---|---|---|---|---|---|---|---|---|---|---|---|---|
| | 0.1 | 0.2 | 0.4 | 0.6 | 0.8 | 1.0 | 0.1 | 0.2 | 0.4 | 0.6 | 0.8 | 1.0 |
| $\epsilon=1$ | 0.0775 | 0.0968 | 0.1145 | 0.1204 | 0.1232 | 0.1249 | 0.1304 | 0.1532 | 0.1706 | 0.1731 | 0.1776 | 0.1782 |
| $\epsilon=2$ | 0.0642 | 0.078 | 0.0983 | 0.1091 | 0.1153 | 0.1196 | 0.1126 | 0.1312 | 0.1543 | 0.1645 | 0.1699 | 0.1732 |
| $\epsilon=5$ | 0.0565 | 0.0634 | 0.0749 | 0.0859 | 0.0959 | 0.1031 | 0.1017 | 0.1089 | 0.125 | 0.1388 | 0.148 | 0.1546 |
| $\epsilon=10$ | 0.0549 | 0.0569 | 0.0631 | 0.0697 | 0.0771 | 0.0837 | 0.1001 | 0.1027 | 0.1091 | 0.1182 | 0.1255 | 0.1321 |

Table 4: $\ell_1$ metric for reconstruction attack with varying $R$ and $\epsilon$

| $\ell_2$ | MNIST | | | | | | UTKFace | | | | | |
|---|---|---|---|---|---|---|---|---|---|---|---|---|
| | 0.1 | 0.2 | 0.4 | 0.6 | 0.8 | 1.0 | 0.1 | 0.2 | 0.4 | 0.6 | 0.8 | 1.0 |
| $\epsilon=1$ | 0.0471 | 0.0613 | 0.0762 | 0.0817 | 0.0847 | 0.0867 | 0.029 | 0.0385 | 0.0464 | 0.0476 | 0.0492 | 0.0498 |
| $\epsilon=2$ | 0.0363 | 0.0478 | 0.0631 | 0.0716 | 0.0773 | 0.0814 | 0.0223 | 0.029 | 0.0387 | 0.0433 | 0.0454 | 0.0471 |
| $\epsilon=5$ | 0.03 | 0.0354 | 0.0449 | 0.0537 | 0.0618 | 0.0672 | 0.0189 | 0.021 | 0.027 | 0.032 | 0.0358 | 0.0388 |
| $\epsilon=10$ | 0.0291 | 0.0305 | 0.0352 | 0.0407 | 0.0467 | 0.0518 | 0.0184 | 0.0192 | 0.0212 | 0.0243 | 0.0271 | 0.0294 |

Table 5: $\ell_2$ metric for reconstruction attack with varying $R$ and $\epsilon$

| PSNR | MNIST | | | | | | UTKFace | | | | | |
|---|---|---|---|---|---|---|---|---|---|---|---|---|
| | 0.1 | 0.2 | 0.4 | 0.6 | 0.8 | 1.0 | 0.1 | 0.2 | 0.4 | 0.6 | 0.8 | 1.0 |
| $\epsilon=1$ | 61.87 | 60.70 | 59.74 | 59.45 | 59.29 | 59.19 | 64.42 | 63.18 | 62.36 | 62.25 | 62.10 | 62.05 |
| $\epsilon=2$ | 63.04 | 61.80 | 60.57 | 60.02 | 59.68 | 59.47 | 65.59 | 64.43 | 63.16 | 62.66 | 62.45 | 62.29 |
| $\epsilon=5$ | 63.88 | 63.15 | 62.09 | 61.28 | 60.67 | 60.29 | 66.30 | 65.84 | 64.75 | 63.99 | 63.50 | 63.15 |
| $\epsilon=10$ | 64.06 | 63.82 | 63.19 | 62.54 | 61.91 | 61.44 | 66.43 | 66.25 | 65.80 | 65.21 | 64.72 | 64.36 |

Table 6: PSNR metric for reconstruction attack with varying $R$ and $\epsilon$

