# OpenReview forum: "Posthoc Privacy guarantees for neural network queries"
_ICLR.cc/2023/Conference — Submitted to ICLR 2023_

### Official Review · Reviewer_KPwN · 2022-10-24

**Confidence:** 4
**Clarity, Quality, Novelty And Reproducibility:** The paper is quite easy to read and t…
**Correctness:** 3
**Technical Novelty And Significance:** 2
**Empirical Novelty And Significance:** 3
**Recommendation:** 6

**Strength And Weaknesses:**

Strengths:
1) The paper presents a novel theoretically motivated approach to provide a privacy guarantee for queries on pre-trained representation learning models.
2) The proposed framework is very similar to DP based approaches and has some properties similar to DP.
3) The mathematical claims of the paper seem correct (based on my reading).

Weakness:
1) Definition of privacy: In Definition 1, semantic neighborhood privacy is also a function of $\theta$. So, it should be included in the definition. This unfortunately significantly weakens this notion of privacy.
2) Identifying appropriate value of R: The authors state "To assess the level of indistinguishability, we look at Fig 3 where
we project the original images into embedding space and sample points from the boundary of
neighborhoods of different R. ". This simply isn't a reasonable approach to select a privacy parameter especially since it relies on humans perception of similarity.
3) The authors don't really propose a practical way to select $\Delta_{LS}^p$.
4) The authors state that the runtime exponentially increases as a function of embedding dimension, which is problematic.
5) Related to weakness 4, the authors evaluate their runtimes on relatively small embedding dimensions. However, there are many use cases when the embedding dimensions are much higher (>1000). What is the feasibility of this approach for such embedding dimensions? I also have significant concerns about how meaningful the privacy guarantees are in the case of high dimensional embeddings.
6) I would be interested in a privacy/utility comparison with DP based model training.

**Summary Of The Paper:**

The paper presents a novel framework for providing post-hoc privacy for queries on models trained using Adversarial Representation Learning (or representation learning in general). The authors draw inspiration on traditional differential privacy (DP) based approaches and are able to provide formal post-hoc privacy guarantees under certain very strong assumptions.

**Summary Of The Review:**

While the proposed approach is somewhat well motivated, I think there are many potential issues with the proposed framework that significantly limit the utility of the framework to some very specific situations (low dimensional embeddings where the embedding space is sufficiently smooth). To their credit, the authors are quite upfront about the limitations of their framework but they overstate the utility in my opinion.

---

> ### Author Response · Authors · 2022-11-07
> **Response to the weaknesses**
>
> We thank the reviewer for their valuable feedback. We generally agree with the drawbacks mentioned and, as acknowledged by the reviewer, we discussed them in the paper. As a main argument, we would like to point out that currently there is no way of giving a privacy guarantee for ARL techniques. Despite some of the current drawbacks, we believe that our proposed framework can be a good first step for evaluating future ARL techniques and further improvements to our framework can improve the utility of this posthoc approach. Now we address the specific concerns -
>
> **W1: Privacy definition is dependent on theta**
> 1. Since our privacy definition is based on metric-DP, the choice of distance metric could be arbitrary as long as it is a well-defined metric.
> 2. We include $\theta$ in the definition intentionally to enforce more meaningful and tangible guarantees. Under our framework, one can replace the $\theta$ based distance metric with $\ell_1$ or $\ell_2$ norms for distance in the ambient dimension. However, such a distance metric will be far from the general intuition of what is expected from reconstruction privacy. For instance - two perceptually similar samples can have a high euclidean distance and similarly two perceptually dissimilar samples can have a low euclidean distance. Therefore, we need a distance metric that captures the similarity between samples at a semantic level. We agree that the privacy guarantees now depend on theta but we believe from a practical standpoint, it is better than giving guarantees in the ambient dimension of data.
>
> **W2: Identifying the appropriate value of R**
> 1. We agree that relying on human perception of similarity may not be a theoretically principled approach. However, in our understanding, the whole task of choosing how much reconstruction leakage is acceptable is inherently tied to human perception. Let us look at how epsilon is chosen in differential privacy - existing works[1,2] have argued that there isn’t any single principle for choosing epsilon value but the choice of epsilon has connections with identifiability which can be chosen based on heuristics such as 1/3 used in Lee and Clifton[2] or econometric cost functions used in Hsu et al[1]. For geolocation privacy[3], a service provider might use privacy within a radius of 500m, however, such absolute units may not exist for images. We believe that from a practical standpoint, choosing a privacy parameter boils down to the service provider and user experience.
> 2. Lemma 2.2 makes a connection between traditional DP and the group size parameter R by showing that a DP algorithm with $\epsilon$ privacy budget is equivalent to $R\epsilon$ privacy budget for a group size of $R$. A similar result holds true for our proposed definition as well. While this equivalence could be used to identify effective epsilon, we note that changing group size makes the notion of neighboring databases different which in turn results in a different privacy guarantee.
> References:
> <sub>[1] Hsu, Justin, Marco Gaboardi, Andreas Haeberlen, Sanjeev Khanna, Arjun Narayan, Benjamin C. Pierce, and Aaron Roth. "Differential privacy: An economic method for choosing epsilon." In 2014 IEEE 27th Computer Security Foundations Symposium, pp. 398-410. IEEE, 2014.
> [2] Lee, Jaewoo, and Chris Clifton. "How much is enough? choosing ε for differential privacy." In International Conference on Information Security, pp. 325-340. Springer, Berlin, Heidelberg, 2011.
>     [3] Andrés, Miguel E., Nicolás E. Bordenabe, Konstantinos Chatzikokolakis, and Catuscia Palamidessi. "Geo-indistinguishability: Differential privacy for location-based systems." In Proceedings of the 2013 ACM SIGSAC conference on Computer & communications security, pp. 901-914. 2013.</sub>
>
> **W3: Selecting $\Delta_{LS}^p$** - The main idea behind using $\Delta_{LS}^p$ instead of global sensitivity is that average local sensitivity is lower than global sensitivity and global sensitivity can be expensive to compute for a neural network. To incorporate this notion of average, we choose $\Delta_{LS}^p$ by first computing the mean and standard deviation of local sensitivity on the training dataset (assumed to be known under our threat model), then we choose mean + n*sigma to have a trade-off between the likelihood of releasing the samples vs adding extra noise to data. We used $n=3$ in our experiments. Since empirically, the value of local sensitivity appears to be following a gaussian, using confidence interval serves as a good proxy. We realized the manuscript was missing detailed information on this and hence we have updated it.
>
> We discuss other weaknesses in the next comment due to the word limit.

---

> ### Author Response · Authors · 2022-11-07
> **Continuation of Response to the weaknesses**
>
> **W4: Exponential relation with embedding dimension** - We agree with this weakness. Since the local Lipschitz constant is an upper bound on the local sensitivity, the exact Lipschitz constant estimator can be replaced by an estimator that gives an upper bound instead but is much faster. All ARL works cited in our paper use Face datasets for benchmarking the privacy-utility trade-off. For such benchmark datasets, we believe that our approach might be sufficient.
>
> **W5: On embedding dimensions being much higher in certain cases** - We respectfully disagree. Under our proposed framework, the goal of the embedder is to go closer to the intrinsic dimension of data and model the image manifold. We refer the reviewer to Figure 1 of Pope et al[1] where it is shown that the intrinsic dimension of image datasets like ImageNet is below 50.
> <sub>[1] Pope, Phillip, Chen Zhu, Ahmed Abdelkader, Micah Goldblum, and Tom Goldstein. "The intrinsic dimension of images and its impact on learning." arXiv preprint arXiv:2104.08894 (2021).</sub>
>
> **W6: Comparison with DP based model training** - DP based model training is for protecting training data and will not impact the privacy of data used for inference. Therefore, our proposed framework is agnostic to the privacy of the training algorithm.
> We do compare with LDP noise injected in the image space and embedding space during the inference stage in Table 2. However, traditional LDP is designed for much stronger guarantees than our proposed definition and can not give any non-trivial privacy-utility trade-off due to stronger privacy assumptions as mentioned in the Preliminaries (Section 2).

---

> ### Author Response · Authors · 2022-11-10
> **Looking forward to discussion**
>
> Dear Reviewer,
> We are looking forward to a productive discussion on the points you mentioned. Please let us know your thoughts on our rebuttal. Thanks for your time.

---

### Official Review · Reviewer_yTGh · 2022-10-25

**Confidence:** 3
**Correctness:** 3
**Technical Novelty And Significance:** 2
**Empirical Novelty And Significance:** 2
**Recommendation:** 3

**Clarity, Quality, Novelty And Reproducibility:**

The contribution and the method is well explained and the choices behind the approach are justified.
The adversarial model is not stated which makes it difficult to understand on who the paper is trying to protect against and at what stage. Who are the parties involved?
It seems that to compute the sensitivity, the obfuscator (i.e., the user) needs to know the classifier. However if it knows it, why would it then send its data for classification to the server in the first place?

It is not clear that hamming, l1 or l2 distance will guarantee privacy. Reconstruction of an image is not the only goal of an attacker.

**Strength And Weaknesses:**

Strength:
- formalising privacy guarantees of obfuscators is an important research problem to avoid erroneous methods
- metric DP and PTR framework build on existing work

Weaknesses:
- adversarial model is not given
- the method relies on knowledge of the classifier which seems at odds with the usefulness of the approach (i.e., why call the classifier)
- metric-DP may not capture all privacy vulnerabilities (i.e., the privacy guarantee is as good as the metric that is used)

**Summary Of The Paper:**

The paper presents a formal way to capture obfuscation of an image using adversarial representation. The paper does so by using a metric-based differential privacy notion. Since the metric is difficult to compute, the paper proposes a way to estimate Lipschitz constant using an estimation technique from DP. The paper then evaluates several obfuscation techniques on their utility.

**Summary Of The Review:**

Given that there were unsuccessful attempts before on adhoc techniques to obtain image privacy by changing/randomizing the enbedding, this paper makes an attempt to provide formal guarantees to capture it. However, the method may not protect against all attacks and seems to rely on the knowledge of the classifier defeating the purpose of sending data for classification.

---

> ### Author Response · Authors · 2022-11-07
> **Rebuttal**
>
> We thank the reviewer for their feedback. We address the two main concerns below -
>
> **On Adversarial model** - We discuss the adversarial model on Page 5, first paragraph. We also discuss our threat model in the Preliminaries (section 2) that motivates our proposed privacy definition. Quoting from the first paragraph on the second page -
> >The scope of our paper is to provide privacy guarantees against reconstruction attacks for existing ARL techniques, i.e., our goal is not to develop a new ARL technique but rather to develop a formal privacy framework compatible with existing ARL techniques. A Majority of the ARL techniques protect either a sensitive attribute or reconstruction of the input. We only consider sensitive input in this work.
>
> We would also like to highlight that ARL techniques considered in this work only focus on preventing reconstruction attacks and that is why we resort to metric-DP.
>
> **On dependence upon the classifier** - We respectfully disagree. Our method does not rely on any knowledge about the ``classifier``. The only model that is required for privatizing data is the obfuscator model. Please let us know if there are any lines in the manuscript that might have implied this and we will update the manuscript accordingly.

---

> ### Author Response · Authors · 2022-11-10
> **Looking forward to discussion**
>
> Dear Reviewer,
> We are looking forward to a productive discussion on the points you mentioned. Please let us know your thoughts on our rebuttal. Thanks for your time.

---

> > ### Comment · Reviewer_yTGh · 2022-12-14
> > **Response**
> >
> > Dear authors, thank you for your response.
> >
> > Adversarial model and threat model: thank you for the pointers the the paper. Unfortunately the threat model does not describe capabilities of the adversary. Hence privacy guarantees cannot be interpreted formally.
> > (E.g. do they know how the obfuscator was trained. If they do, it may reveal to them more about the image. If not, this is a strong assumption to make and has to be clearly stated. How many times can the adversary interact with an obfuscator? Is the adversary adaptive. If they were to see (input,ouput) pairs of the obfuscator and then given an obfuscated image, would they be able to infer what this image was. These assumptions have to be clearly stated. That is if it is not a DP guarantee that is provided, then a cryptographic definition should be in place (e.g., see semantically secure encryption).)
> >
> > Privacy guarantees: reconstruction attacks are not the only attacks that are possible. Hence, it will be ambiguous to say that this method provides privacy or privacy-preserving encoding. For example, it seems attacks described in "Is Private Learning Possible with Instance Encoding?" by Carlini et al. are still possible as instance encoding is not possible unless one uses encryption. Though this work is referenced, it is not stated whether these attacks will not be possible using the method in the paper. As a result, I believe this work will create ambiguity of the guarantees it can actually provide (together with unclear threat model) and maybe susceptible to information leakage as described in the work by Carlini et al..
> >
> > Adversarial training for privacy: this method has been shown to not guarantee privacy in general as one cannot solve the optimization problem precisely and relaxations are needed. (e.g., see "Overlearning Reveals Sensitive Attributes", ICLR 2020).
> >
> > Classifier: Thank you for clarifying this that a class to classifier is indeed required to do an inference.

---

### Official Review · Reviewer_82bi · 2022-10-27

**Confidence:** 4
**Correctness:** 3
**Technical Novelty And Significance:** 3
**Empirical Novelty And Significance:** 2
**Recommendation:** 6

**Clarity, Quality, Novelty And Reproducibility:**

The writing in the experiments part is not good, the Tables and Figures are not explained enough.
The method seems novel, though simple.

**Strength And Weaknesses:**

Overall, I think this is a good paper with decent results, though the core of paper is a rather simple idea of relating Lipschitz constant to local sensitivity, other techniques are mostly borrowed from other papers.
There are terms used in Alg 1, which are defined after the algorithm, as as N. I feel the part after the algorithm can be stated first and then the algorithm.
Fig. 2 could have better legend for the colors for R.
I do not understand what the epsilon values are on the very first heading in Table 1
What is the encoder row?

**Summary Of The Paper:**

The paper utilizes a method called Adversarial Representation Learning (ARL) as a base. ARL
learns a privacy-preserving encoding of sensitive user data before it is shared, but lacks formal guarantees. The authors link the local Lipschitz constant of a neural network in ARL (obfuscation layer) with its local sensitivity. Local sensitivity was used by the Propose-Test-
Release (PTR) framework to provide privacy guarantee, which the authors modify to work with the Lipschitz constant.
Experimentally on datasets verify the role of ARL in improving the privacy-utility tradeoff.

**Summary Of The Review:**

Good work, though simple extension. The writing can be better.

---

> ### Author Response · Authors · 2022-11-07
> **Rebuttal**
>
> We thank the reviewer for their feedback. We address the main concerns here -
>
> **On novelty** - We propose a formal solution to a privacy problem for which only empirical techniques currently exist. We also empirically validate the practical efficacy of our formal framework. There are three technical novel ideas in this paper -
> - N1: Proposing a privacy definition that captures reconstruction in the context of Adversarial Representation Learning.
> - N2: Linking Lipschitz constant to local sensitivity.
> - N3: Proposing a variant of Propose-Test-Release, since PTR can not be directly applied due to computational tractability. This variant also requires new proofs for privacy guarantee which led to Lemma 4.2 and 4.3.
>
> We believe that N3 is the main technical novel contribution among all things considered.
>
> In addition, we also empirically identify the role of ARL in improving the privacy-utility trade-off by showing that adversarial regularization reduces the Lipschitz constant which is related to local sensitivity. To the best of our knowledge, this result between ARL and Lipschitz constant has not been shown before and can be even useful for other research areas in deep neural networks.
>
> **On writing** -  Thanks for your feedback on the writing aspect, we agree with the points mentioned and hence have improved the manuscript to address the concerns.
>
> - We have moved the algorithm section to the Supplementary and referenced in the end of the method section.
> - We have added more details about the experiments and improved clarification. This includes adding more details explaining the results shown in Tables and Figures.

---

> ### Author Response · Authors · 2022-11-10
> **Looking forward to discussion**
>
> Dear Reviewer,
> We are looking forward to a productive discussion on the points you mentioned. Please let us know your thoughts on our rebuttal. Thanks for your time.

---

### Author Response · Authors · 2022-11-18
**Summary of our response to the weaknesses mentioned by the reviewers**

While we have responded to each criticism inline, here we identify the main ones

``Reviewer 82bi``: **On Novelty**: We believe the first three contributions mentioned above make a case for novelty. While our ideas build upon existing works, they could not be integrated trivially and required novel contributions in some instances which we discuss in our inline response.

``Reviewer yTGh``: **On the lack of an Adversarial model**: We described the adversarial setup in detail in the second paragraph of Section 4. We also discuss the threat model in the Preliminaries section to motivate why we need a different definition than traditional DP.

``Reviewer KPwN``: We generally agree with the reviewer's summary that our framework is effective for datasets with low-dimensional and smooth embedding space. Experimentally, we embedded our data using a Variational Autoencoder. We believe the usage of such an embedder is applicable to several datasets where the generative ML community already uses latent space interpolation.

* **Privacy definition dependent on $\theta$**: Unlike traditional DP that computes the distance between datasets in the hamming space, we use $\ell_1$ distance in the embedding space instead. Our privacy definition, like traditional DP, is a specific instantiation of metric-DP[2], and therefore the choice of metric leads to different privacy guarantees. We use $\ell_1$ distance in the embedding space because measuring $\ell_1$ distance in the ambient space (pixel space for images) does not give useful guarantees for distributions like images that lie on a lower dimensional manifold[1].

* **Choosing privacy parameter $R$**: Choosing privacy parameters is generally a difficult problem in data privacy. The parameter $R$ introduced in our work is analogous to the group size used in traditional DP which is kept 1 by default in order to guarantee privacy at an individual level. There is an equivalence relation between the $\epsilon$ of group privacy and individual privacy (Lemma 2.2). However, when the goal is to prevent reconstruction (as done in ARL) choosing $R$ will depend on the application based on how much indistinguishability in reconstruction we would like to guarantee for an attacker. Our empirical reconstruction attack results show this trade-off.

* **Proposing $\Delta_{LS}^p$**: We have added more details both in the rebuttal and the manuscript about a principled way to do this.

* **Runtime**: Since estimating local Lipschitz constant requires solving a mixed integer program, the runtime for our algorithm increases exponentially with the input dimension. However, the input dimension in our setup refers to the embedding dimension which is pretty low because dimensions for latent space are much lower than ambient data space for image datasets. This aspect we believe can be further improved in future works by potentially re-using optimization results. Furthermore, as we show in our experiments, for synthetic datasets (MNIST and FMNIST) and a real face image dataset (UTKFace), we can apply our framework at almost a real-time speed (2-3 sec/image). This is due to the design choices mentioned in Section 4 last paragraph on computation feasibility.

* **DP-based model training comparison**: A DP-based model training would not provide privacy guarantees for the data used during inference.


References

1. Pope, Phillip, Chen Zhu, Ahmed Abdelkader, Micah Goldblum, and Tom Goldstein. "The intrinsic dimension of images and its impact on learning." arXiv preprint arXiv:2104.08894 (2021).

---

### Author Response · Authors · 2022-11-18
**Author summary**

We highlight the main aspects of this paper and our thoughts on the criticisms.

We propose a formal framework for neural network (NN) based obfuscation techniques. The posthoc nature of our framework allows reasoning about privacy without any modification to the obfuscation algorithm itself making it easy to integrate for benchmarking existing and future techniques. To the best of our knowledge, currently, there isn’t any way of giving formal guarantees for NN based obfuscation techniques. The problem is challenging because neural networks used for obfuscation are non-linear and optimized over non-convex objective functions.
We have attempted to solve the problem by making the following contributions -

*     1. Proposing a privacy definition that formalizes reconstruction privacy in the context of single-instance encoding which is what obfuscation techniques aim to protect. Protecting against membership inference (meaningfully) is not possible under this setup and hence traditional DP can not be applied directly. (see - Section 2 of our paper and a research paper on the impossibility of instance encoding[1]).

*     2. Designing a variant of the Propose-Test-Release (PTR) algorithm that guarantees privacy using the local Lipschitz constant. We note that the standard PTR can not be applied trivially due to its intractability for queries beyond median and mode (see - Page 152, 2nd paragraph of Dwork and Roth[2] and Page 6, 1st paragraph of our paper). Designing this variant entails -
       a) Linking the local Lipschitz constant of a neural network with its local sensitivity under our proposed definition. (Lemma 4.1)
       b) Deriving a tractable lower bound on a query used in PTR that is intractable to compute for functions such as neural networks. (Lemma 4.2)
       c) A binary search algorithm to obtain this lower bound tractably by integrating it with a Lipschitz constant estimator. (Algorithm 1)

*     3. Experiments 1) to verify efficacy, 2) ablation on different design choices, 3) empirical reconstruction attacks, and 4) identifying the role Adversarial techniques play in improving privacy-utility trade-off.

References
1. Carlini, Nicholas, Samuel Deng, Sanjam Garg, Somesh Jha, Saeed Mahloujifar, Mohammad Mahmoody, Shuang Song, Abhradeep Thakurta, and Florian Tramer. "An Attack on InstaHide: Is Private Learning Possible with Instance Encoding?." (2020).
2. Dwork, Cynthia, and Aaron Roth. "The algorithmic foundations of differential privacy." Foundations and Trends® in Theoretical Computer Science 9, no. 3–4 (2014): 211-407.

---

### Decision · Program_Chairs · 2023-01-20

**Decision:**

Reject

**Justification For Why Not Higher Score:**

The theoretical setting and privacy definition is rather narrow. The privacy definition is not clear enough.

**Justification For Why Not Lower Score:**

N/A

**Metareview: Summary, Strengths And Weaknesses:**

This paper proposes a narrow privacy definition, geared towards single-user responses. It leverages recent adversarial techniques and proposes a post-hoc analysis based on local sensitivity and propose test release. Although one reviewer was very negative about the paper, it seems that this partially stemmed from a misunderstanding.

Nevertheless, subsequent discussion showed that there are still some things that need to be clarified or made more precise, and in particular the threat model: It seems that the privacy guarantees do rely on the adversary not having access to the obfuscation mechanism, but this should be made clear in the paper, at the very least. The privacy implications of those assumptions are unclear, but they could make the construction weak overall.



**Summary Of Ac-Reviewer Meeting:**

We didn't have a meeting, I just read their comments and responses.